# Unifying Generative Models with GFlowNets and Beyond

## Abstract

There are many frameworks for generative modeling, each often presented with their own specific training algorithms and inference methods. Here, we demonstrate the connections between existing generative models and the recently introduced GFlowNet framework (Bengio et al., 2021b), a probabilistic inference machine which treats sampling as a decision-making process. This analysis illuminates the common characteristics they share and offers a new perspective by considering them in the context of learning with Markovian trajectories. Our framework provides a route to shine a unifying light over many (but not all) generative models. Beyond this, we provide a practical and experimentally verified recipe for improving generative modeling with insights from the GFlowNet perspective.

## 1 Introduction

Generative models are a class of machine learning algorithms that use probabilistic methods to capture and perform inference over complex distributions, usually from a given training dataset. They have a wide range of applications, including data generation, anomaly detection, probabilistic inference and density estimation. In the past few decades, a variety of different generative models have been developed, each with its own set of assumptions and capabilities.

Early examples of generative models include probabilistic graphical models, such as Bayesian networks and Markov random fields (Koller & Friedman, 2009; Murphy, 2012), and latent variable models, such as latent Dirichlet allocation (Blei et al., 2001) and Helmholtz machines (Dayan et al., 1995). These models have proven to be effective at capturing dependencies within data.

The research on generative modeling has taken off during the past decade, thanks to the representational power of deep neural networks. One well-known example is the generative adversarial network or GAN for short (Goodfellow et al., 2014), which consists of a generator network that stochastically produces new samples and a deterministic discriminator network that tries to distinguish between real and generated samples. Another popular method is the variational autoencoder (Kingma & Welling, 2013), which learns a hierarchical latent variable model to express the target distribution with the help of a variational posterior. Other types of generative models based on deep networks (deep generative models) include: normalizing flows (Dinh et al., 2014), which transform a simple distribution into a target distribution via a series of invertible transformations; autoregressive (AR) models (Bengio & Bengio, 1999; van den Oord et al., 2016), which model the data by decomposing it into a product of conditional distributions; and energy-based models (Hinton, 2002; LeCun et al., 2006), which models the negative log probability of a distribution. Recently, denoising diffusion models (Vincent, 2011; Sohl-Dickstein et al., 2015; Ho et al., 2020; Song et al., 2020) have shown impressive results in generating high quality samples. Their modeling could be seen as a series of denoising steps which gradually transform white noise into noise-free data by stochastically inverting the process of transforming real data into white noise through a sequence of noise injection steps. Each of these generative models has its own set of assumptions and limitations, which can make it challenging to choose the right model for a particular task (Hu et al., 2018).

GFlowNets (Bengio et al., 2021a;b), short for generative flow networks, is a class of samplers which stems from a reinforcement learning or RL for short (Sutton & Barto, 2005) formulation. GFlowNets treat the sampling process as a sequential decision-making process, and learn a stochastic (forward) policy to sample

compositional objects with probability proportional to a given terminating reward function. It has been demonstrated that GFlowNets are able to sample from diverse modes rather than being stuck in single modes like is typical of Markov chain Monte Carlo (MCMC) or variational inference methods (Zhang et al., 2022b; Malkin et al., 2023), which is of great importance in drug discovery (Jain et al., 2022a; Zhang et al., 2021).

In this paper, we show how some of existing generative models could be taken as special cases of GFlowNets, with their modeling components specified in different probabilistic ways. We thus propose to treat the GFlowNet as a probabilistic framework for unifying many different kinds of generative models, and facilitate the analysis of their connections and possible extensions (Section 3). Further, after analysing the relationship between GFlowNet setup and generative modeling setup in Section 2.3, we propose MLE-GFN, a generative modeling algorithm inspired by GFlowNet ideas in Section 4. The proposed algorithm improves the performances of existing generative modeling baselines on both discrete and continuous image modeling tasks.

## 2 Preliminaries and Notations

### 2.1 Generative Modeling

Generative modeling aims to use probabilistic methods to model a distribution from a given dataset $\mathfrak{D} = \{\mathbf{x}_i\}_i, \mathbf{x}_i \in \mathcal{X}$, where $\mathcal{X}$ is the space of data objects. During training, we want to learn a distribution $q(\mathbf{x})$ to be close to the target distribution $p^*(\mathbf{x}) = \frac{1}{|\mathfrak{D}|} \sum_i \delta_{\{\mathbf{x}_i - \mathbf{x}\}}$, where $\delta_{\{\cdot\}}$ is the Dirac delta function. However, the real objective is to generalize well to an underlying data generating process $p^*$ from which both training samples and test samples may be drawn. One example to achieve this is to minimize the KL divergence, *i.e.*, $\min_q \mathcal{D}_{\mathrm{KL}}(p^* \| q)$, which corresponds to the maximum likelihood estimation (MLE), a popular method for training generative models. Other methods could also be taken as examples of divergence minimization, *e.g.*, GAN's adversarial training could be viewed as minimizing the Jensen-Shannon divergence between the model and the target distribution.

### 2.2 GFlowNets

From a probabilistic modeling viewpoint, a generative flow network (GFlowNet) (Bengio et al., 2021a;b) is a probabilistic inference methodology that aims to sample $\mathbf{x} \in \mathcal{X}$ in proportion to a given reward function $R(\mathbf{x})$, where $\mathcal{X}$ is the set of data. That is to say, the target distribution $p^*(\cdot)$ that we want so sample from satisfies $p^*(\mathbf{x}) \propto R(\mathbf{x})$. Recently, the community has experienced a progressive expansion of the concept of GFlowNets (Malkin et al., 2022; Deleu et al., 2022; Jain et al., 2022a;b; Pan et al., 2022; Madan et al., 2022; Liu et al., 2022; Lahlou et al., 2023b).

More precisely, a GFlowNet samples a Markovian trajectory $\tau = (\mathbf{s}_0, \mathbf{s}_1, \ldots, \mathbf{s}_n)$ with length $n+1$, where $\mathbf{s}_i \in \mathcal{S}$ is the intermediate states for all $i$ in $[n] := \{0, 1, \ldots, n\}$, whose space $\mathcal{S}$ is not necessarily the same as $\mathcal{X} \subseteq \mathcal{S}$. If not specially specified, we use the notation $\mathbf{x} = \mathbf{s}_n$ for the final / terminating state of the trajectory. This process has a natural connection to reinforcement learning (Sutton & Barto, 2005; Bengio et al., 2021a; Zhang et al., 2022a), which frames the generation of objects $\mathbf{x}$ as a sequential decision-making problem on a deterministic Markov decision process (MDP) with set of actions being $\mathcal{A} \subseteq \mathcal{S} \times \mathcal{S}$. The set of all trajectories $\tau$ form a directed acyclic graph (DAG) in the latent state space, whose nodes are states $\mathbf{s} \in \mathcal{S}$. Each complete trajectory starts from the same (abstract) initial state $\mathbf{s}_0$ and ends in a terminating state $\mathbf{s}_n$. The (trajectory) flow function $F(\tau) \in \mathbb{R}_+$ defined by Bengio et al. (2021b) can be understood by an analogy with the number of water particles flowing through trajectory $\tau$ in a network of pipes with $s_0$ as single source and all the terminating states as sinks. Ideally, we want the amount of flow leading to $\mathbf{x}$ equals to the given reward: $\sum_{\tau=(\mathbf{s}_0,\cdots,\mathbf{s}_n),\mathbf{s}_n=\mathbf{x}} F(\tau) = R(\mathbf{x})$. The GFlowNet state space $\mathcal{S}$ could be either discrete or continuous, as demonstrated in Lahlou et al. (2023b).

The forward policy on the GFlowNet MDP is a conditional distribution $P_F(\mathbf{s}' \mid \mathbf{s})$ over all child states $\mathbf{s}'$ that are reachable from $\mathbf{s}$ within one transition. A distribution over trajectories $P_F(\tau)$ can be induced by the

forward policy, whose definition is

$$P_F\left(\tau = (\mathbf{s}_0 \rightarrow \mathbf{s}_1 \rightarrow \ldots \rightarrow \mathbf{s}_n)\right) = \prod_{i=0}^{n-1} P_F\left(\mathbf{s}_{i+1}|\mathbf{s}_i\right). \tag{1}$$

We want to learn the GFlowNet policy such that the distribution over GFlowNet generated objects $P_F^\top(\mathbf{x}) = \sum_{\tau \rightarrow \mathbf{x}} P_F(\tau)$ is the same as the target distribution $p^*(\mathbf{x}) \propto R(\mathbf{x})$. Here $\top$ denotes terminating. To determine a GFlowNet, it suffices to specify its forward policy (Bengio et al., 2021b).

We then introduce several GFlowNet training criterions. The GFlowNet **detailed balance** (DB) constraint writes $F(\mathbf{s})P_F\left(\mathbf{s}' \mid \mathbf{s}\right) = F(\mathbf{s}')P_B\left(\mathbf{s} \mid \mathbf{s}'\right)$, where $P_F(\mathbf{s}'|\mathbf{s})$ and $P_B(\mathbf{s}|\mathbf{s}')$ are the forward and backward policy respectively and characterize the stochastic transitions between different states, going forward or backward along a trajectory. Here the backward policy is a conditional distribution over all parent states $\mathbf{s}$ of a given state $\mathbf{s}'$, and $F(\mathbf{s})$ is the state flow function which is related to the summation of all trajectory flows through $\mathbf{s}$, *i.e.*, $F(\mathbf{s}) = \sum_{\tau \ni \mathbf{s}} F(\tau)$. For a practical deep learning algorithm with DB, we can separately parameterize three models for $F_{\boldsymbol{\theta}}(\cdot), P_F(\cdot|\cdot; \boldsymbol{\theta}), P_B(\cdot|\cdot; \boldsymbol{\theta})$ and updating them by minimizing the squared loss objective $(\log F(\mathbf{s}; \boldsymbol{\theta}) + \log P_F(\mathbf{s}' \mid \mathbf{s}; \boldsymbol{\theta}) - \log F(\mathbf{s}'; \boldsymbol{\theta}) - \log P_B(\mathbf{s} \mid \mathbf{s}'; \boldsymbol{\theta}))^2$.

Extending the detailed balance criterion to a constraint on the whole trajectory, the **trajectory balance** criterion (Malkin et al., 2022) aims to match GFlowNet's forward trajectory probability $P_F(\tau)$ and the backward trajectory probability $P_B(\tau)$ via $\mathcal{L}_{\text{TB}}(\tau; \boldsymbol{\theta}) = (\log P_F(\tau; \boldsymbol{\theta}) - \log P_B(\tau; \boldsymbol{\theta}))^2$, where

$$P_F(\tau; \boldsymbol{\theta}) = \prod_{i=0}^{n-1} P_F\left(\mathbf{s}_{i+1}|\mathbf{s}_i; \boldsymbol{\theta}\right), \quad P_B(\tau; \boldsymbol{\theta}) = \frac{R(\mathbf{x})}{Z_{\boldsymbol{\theta}}} \prod_{t=0}^{n-1} P_B\left(\mathbf{s}_i|\mathbf{s}_{i+1}; \boldsymbol{\theta}\right), \tag{2}$$

and $\mathbf{x} = \mathbf{s}_n$ which is the terminating state in $\tau$. Here $Z_{\boldsymbol{\theta}}$ is a learnable scalar parameter, which we aim to use to approximate the normalizing factor $\sum_{\mathbf{x}} R(\mathbf{x})$. We also use the notation of $P_B(\tau|\mathbf{x}) = \prod_{i=0}^{n-1} P_B\left(\mathbf{s}_i \mid \mathbf{s}_{i+1}\right)$ when $\mathbf{x}$ is the terminating state of $\tau$. Malkin et al. (2022) proposes to use such square loss training objective; nonetheless, in this work we also focus on general divergences between the two distributions; *e.g.*, with Kullback–Leibler (KL) divergence, we call such specification of $\mathcal{L}_{\text{TB}}(\tau; \boldsymbol{\theta}) = \mathbb{E}_{P_F}\left[\log P_F(\tau; \boldsymbol{\theta}) - \log P_B(\tau; \boldsymbol{\theta})\right]$ or $\mathcal{L}_{\text{TB}}(\tau; \boldsymbol{\theta}) = \mathbb{E}_{P_B}\left[\log P_B(\tau; \boldsymbol{\theta}) - \log P_F(\tau; \boldsymbol{\theta})\right]$ to be KL-trajectory balance.

### 2.3 Two Settings: Generative Modeling with Data and Sampling with Given Densities

As would be shown in the following section, the similarity between GFlowNets and generative models mainly exists on the modeling side. As a matter of fact, GFlowNets in its origin are designed for sampling from given unnormalized densities (which we call sampling for short), which is a very different problem setup from generative modeling with given data (which we call generative modeling for short). In the former setting, a training dataset is provided and is treated as an empirical approximation of the target distribution. However, in the latter problems where GFlowNets are proposed to be used, the practitioners are given a black-box (probably unnormalized) target density function $p^*(\cdot)$ as a callable oracle instead of a dataset. That means that in sampling, the learning signal comes from a non-differentiable function which takes in a data point $\mathbf{x}$ and returns a scalar $p^*(\mathbf{x})$. In theory, the probability density function of the target distribution contains all the information about the distribution, and it would be possible to use MCMC sampling to draw a dataset from this density function. Nonetheless, in practice it is computationally infeasible to explore all the modes of its landscape (due to the high dimensionality), especially when the distribution is not unimodal, let alone the MCMC algorithms would take infinite computation time to mix. From this perspective, both generative modeling and sampling aim at learning probabilistic models to represent target distributions, but the former is an easier distribution matching problem than the latter: in generative modeling, the exploration part of sampling has been done, and we only need to exploit the information in the dataset by fitting the generative models.

Under the context of reinforcement learning, the connection between generative modeling and sampling is similar to the relationship between offline RL (Lin, 2004; Lange et al., 2012) and online RL, where in offline RL we are given a dataset of labelled trajectories obtained from the interaction between a predefined expert

agent and a particular environment. It is well known that in online RL, due to the high complexity of the settings, algorithms would give results with high variance (Henderson et al., 2017). Besides, works have shown that the performance of offline RL tasks are much more stable than their online variants (Agarwal et al., 2020). This originates from the fact that in online RL the agent need to interact with the environment to explore the landscape of the task, which is full of uncertainty. The exploration is also a serious challenge that sampling would face. On the other hand, in offline RL the goal is simply to learn an optimal policy from existing data, which is related to imitation learning – generative modeling in the trajectory level.

## 3 GFlowNet as a Unifying Framework

### 3.1 Hierarchical Variational Autoencoders

The evidence lower bound for bottom-up hierarchical VAEs (HVAEs) (Ranganath et al., 2016) reads

$$\log p(\mathbf{x}) \geq \ \text{ELBO}_{p,q}(\mathbf{x}) \triangleq \mathbb{E}_{q(\mathbf{z}_{1:n-1}|\mathbf{x})} \left[ \log p(\mathbf{x}, \mathbf{z}_{1:n-1}) - \log q(\mathbf{z}_{1:n-1}|\mathbf{x}) \right] \tag{3}$$

$$= \mathop{\mathbb{E}}_{q(\mathbf{z}_{n-1}|\mathbf{z}_n)\cdots q(\mathbf{z}_1|\mathbf{z}_2)} \left[ \log p(\mathbf{z}_1) + \sum_{i=1}^{n-1} \log \frac{p(\mathbf{z}_{i+1}|\mathbf{z}_i)}{q(\mathbf{z}_i|\mathbf{z}_{i+1})} \right], \tag{4}$$

where we denote $\mathbf{x} := \mathbf{z}_n$, $\mathbf{z}_{1:n-1} = (\mathbf{z}_1, \ldots, \mathbf{z}_{n-1})$, and $p, q$ respectively denote the hierarchical decoder and encoder of the HVAE. It is well known that this hierarchical ELBO can also be represented as $\log p(\mathbf{x}) - \mathcal{D}_{\text{KL}} \left( q(\mathbf{z}_{1:n-1}|\mathbf{x}) \| p(\mathbf{z}_{1:n-1}|\mathbf{x}) \right)$, where $p(\mathbf{z}_{1:n-1}|\mathbf{x}) \propto p(\mathbf{x}|\mathbf{z}_{1:n-1}) p(\mathbf{z}_{1:n-1})$. As we show below, with a GFlowNet that samples $\mathbf{z}$ given $\mathbf{x}$ and where the reward is $p(\mathbf{x}, \mathbf{z}_{1:n-1})$, we also aim to match the forward trajectory policy which ends with data $\mathbf{x}$ with the corresponding backward trajectory policy, *i.e.*, $P_F(\tau) \approx P_B(\tau)$, conditioning on the event $\{\mathbf{x} \in \tau\}$, *i.e.*, $\mathbf{x}$ is the terminating state of $\tau$. Note that we have $\mathbb{P}(\mathbf{x}|\tau) = \delta_{\{\mathbf{x} \in \tau\}}$, where $\mathbb{P}(\cdot)$ denotes the probability of some event.

**Observation 1.** *The HVAE is a special kind of GFlowNet in the following sense: each trajectory is of the form of $\tau = (\mathbf{z}_0, \mathbf{z}_1, \cdots, \mathbf{z}_n = \mathbf{x})$. The VAE decoder, which samples $\mathbf{z}_0 \to \cdots \to \mathbf{z}_{n-1} \to \mathbf{z}_n$, corresponds to the GFlowNet forward policy; the encoder samples $\mathbf{x} \to \mathbf{z}_{n-1} \to \cdots \to \mathbf{z}_0$, and corresponds to the GFlowNet backward policy.*

With $\mathbf{x} = \mathbf{z}_n$, we could then write

$$P_F(\tau) = p(\mathbf{z}_0) \prod_{i=1}^{n-1} p(\mathbf{z}_{i+1}|\mathbf{z}_i), \tag{5}$$

$$P_B(\tau) = p^*(\mathbf{x}) \prod_{i=1}^{n-1} q(\mathbf{z}_i|\mathbf{z}_{i+1}), \tag{6}$$

where $p^*(\mathbf{x}) = \frac{R(\mathbf{x})}{Z}$ is the true target density, $p(\cdot|\cdot)$ is the forward policy, and $q(\cdot|\cdot)$ is the backward policy. We can see that *the $i$-th state in a GFlowNet trajectory (*i.e.*, $\mathbf{s}_i$) corresponds to $\mathbf{z}_i$*.

The following proposition reveals an equivalence between the two perspectives in Observation 1.

**Proposition 2.** *Training hierarchical latent variable models with the KL-trajectory balance $\mathcal{D}_{\text{KL}}(P_B(\tau)\|P_F(\tau))$ objective is equivalent to training HVAEs by maximizing its ELBO, in the sense of having the same global optimum.*

In this work we relegate all proofs to Section B. We refer to Sønderby et al. (2016); Child (2021); Shu & Ermon (2022) for practice of HVAEs on large scale image modeling.

### 3.2 Diffusion Models

#### 3.2.1 Denoising diffusion probabilistic models

The success of deep learning relies on careful design of inductive biases in the learning algorithm (Goyal & Bengio, 2020). With the same amount of computational resources, the better assumptions we use to

constrain the model, the more powerful the algorithm will be. One way to bake inductive biases into the aforementioned hierarchical VAE model is by forcing the following Gaussian assumptions:

$$P_F(\mathbf{s}_{i+1}|\mathbf{s}_i) = \mathcal{N}(\mathbf{s}_{i+1}; \boldsymbol{\mu}_i(\mathbf{s}_i), \beta_i \mathbf{I}), \tag{7}$$

$$P_B(\mathbf{s}_i|\mathbf{s}_{i+1}) = \mathcal{N}(\mathbf{s}_i; \sqrt{1-\beta_i}\mathbf{s}_{i+1}, \beta_i \mathbf{I}), \tag{8}$$

where $i = 0, \ldots, n-1$ are integer indices, $\mathcal{N}$ denotes the Gaussian distribution, $\{\mu_i(\cdot)\}_i$ are functions to be learned, and $\{\beta_i\}_i$ are constant positive real numbers. In this way, the latent variables at every stage of the sequential generative process share the same number of dimensions as data $\mathbf{x}$.

**Observation 3.** *The denoising diffusion probabilistic model (Ho et al., 2020, DDPM) is a special kind of GFlowNet with the forward / backward policy[1] specified as in Eq. 7 / Eq. 8.*

Remark that our notation of the time index is in the reverse ordering of the one used in the DDPM exposition, which results in a slightly different definition of $\beta$. With the special design in Eq. 7 and 8, a DDPM enjoys an efficient training procedure (each $\boldsymbol{\mu}_i$ can be trained locally by only trying to invert the noise added to $s_{i+1}$ to obtain $s_i$), which helps it scale well and made it become a state-of-the-art method for high-dimensional image generative modeling (Dhariwal & Nichol, 2021; Kingma et al., 2021). We analyze the relationship between its training objective and a GFlowNet formulation in the following proposition.

**Proposition 4** (informal). *Training a GFlowNet defined as in Eq. 7 and 8 with KL-trajectory balance is equivalent to training a DDPM with its regression-based denoising objective.*

**Discrete space diffusion** The above modeling method could also generalize to structured discrete data. For categorical (*i.e.*, discrete) data, $\mathbf{s}_i$ denotes a one-hot representation of the state, and we define a hierarchical model as in Section 3.1 in the manner of the following claim, which parallels Observation 3:

**Observation 5.** *The discrete denoised diffusion model for categorical data proposed by Hoogeboom et al. (2021); Austin et al. (2021) is a special case of GFlowNet with the forward policy specified in Eq. 9 and backward policy specified in Eq. 10.*

$$P_F(\mathbf{s}_{i+1}|\mathbf{s}_i) = \mathcal{C}(\mathbf{s}_{i+1}; \mathbf{p} = \boldsymbol{\mu}_i(\mathbf{s}_i)), \tag{9}$$

$$P_B(\mathbf{s}_i|\mathbf{s}_{i+1}) = \mathcal{C}(\mathbf{s}_i; \mathbf{p} = \mathbf{s}_{i+1}\mathbf{Q}_i), \tag{10}$$

*where $\mathcal{C}$ denotes a categorical distribution, $\{\mathbf{Q}_i\}_i$ are doubly-stochastic constant Markov transition matrices, and $\{\boldsymbol{\mu}_i(\cdot)\}_i$ are parametric functions serving as the categorical parameter for the GFlowNet forward policy.*

### 3.2.2 Denoised diffusion through SDEs

The behavior of a deep latent variable model in its infinite depth regime is studied by Tzen & Raginsky (2019); in the language of GFlowNets, the forward and backward policy take the following form:

$$P_F(\mathbf{s}_{i+1}|\mathbf{s}_i) = \mathcal{N}\left(\mathbf{s}_{i+1}; \mathbf{s}_i + h\mathbf{f}_i(\mathbf{s}_i), hg_i^2\right), \tag{11}$$

$$P_B(\mathbf{s}_i|\mathbf{s}_{i+1}) = \mathcal{N}(\mathbf{s}_i; \mathbf{s}_{i+1} + h(-\mathbf{f}_{i+1}(\mathbf{s}_{i+1}) + g_{i+1}^2 \nabla \log F(\mathbf{s}_{i+1})), hg_{i+1}^2), \tag{12}$$

where we assume all $\mathbf{s}_i$ have the same number of dimensions as $\mathbf{x}$, time step $h = 1/L$, $\{g_i\}_i$ are scalar parameters, and $\{\mathbf{f}_i(\cdot): \mathcal{S} \to \mathcal{S}\}_i$ are parametric mappings. The hierarchical model is then equivalent to a stochastic process in its diffusion limit ($h \to 0$). Huang et al. (2021); Kingma et al. (2021); Song et al. (2021) also study connections between hierarchical variational inference on deep latent variable models and diffusion processes.

Consider a stochastic differential equation (SDE) (Øksendal, 1985) and its reverse time SDE (Anderson, 1982)

$$d\mathbf{x} = \mathbf{f}(\mathbf{x}) \, dt + g(t) \, d\mathbf{w}_t, \tag{13}$$

$$d\bar{\mathbf{x}} = \left[\mathbf{f}(\bar{\mathbf{x}}) - g^2(t)\nabla_{\bar{\mathbf{x}}} \log p_t(\bar{\mathbf{x}})\right] d\bar{t} + g(t) \, d\bar{\mathbf{w}}_t, \tag{14}$$

---

[1]There is another specification of the DDPM generative decoder variance, which we ignore as it does not affect our discussion.

where $\mathbf{f}(\cdot)$ and $g(\cdot)$ are given and $\mathbf{x}, \mathbf{w} \in \mathbb{R}^D$, $\mathrm{d}\mathbf{w}_t$ is a Wiener process, and $\bar{\mathbf{x}}, \bar{t}, \bar{\mathbf{w}}_t$ denote the reverse time version of $\mathbf{x}, t, \mathbf{w}_t$. We define $\mathbb{P}_h(\mathbf{x}_{t+h}|\mathbf{x}_t)$ to be the transition kernel induced by the SDE in Eq. 13, namely $\mathbf{x}_{t+h} = \mathbf{x}_t + \int_t^{t+h} f(\mathbf{x}_\tau) \, \mathrm{d}\tau + \int_t^{t+h} g(\tau) \, \mathrm{d}\mathbf{w}_\tau$, where $h$ denotes an infinitesimal time step. This modeling, adopted and popularized by Song et al. (2020, ScoreSDE), could be connected to GFlowNets as follows.

**Observation 6.** *ScoreSDE is a special case of GFlowNets, in the sense that GFlowNet states take the time-augmented form $\mathbf{s}_t = (\mathbf{x}_t, t)$ for some $t \in [0, 1]$, the SDE in Eq. 13 models the forward policy of GFlowNets (i.e., how states should move forward) while the reverse time SDE in Eq. 14 models the backward policy of GFlowNets (i.e., how states should move backward). In this case, a trajectory $\{\mathbf{s}_t\}_{t=0}^1$ is also in the form of $\{\mathbf{x}_t\}_{t=0}^1$.*

Note that we cannot directly treat $\mathbf{x}_t$ as a GFlowNet state, as the GFlowNets theory requires the graph of all latent states to be a DAG (*i.e.*, one cannot return to an already visited state). This state augmenting operation (Bengio et al., 2021b) induces the required DAGness. This follows because any $(\mathbf{x}, t) \to (\mathbf{x}', t')$ transition with $t \geq t'$ is forbidden. Without loss of generality, we assume that the notation $\mathbf{x}_t$ as a GFlowNet state already contains the time stamp itself in the context below[2].

We now point out an analogy between a stochastic processes property and a GFlowNet property:

**Observation 7.** *The property of a stochastic process*

$$\int p(\mathbf{x}_{t-h}, t-h)\mathbb{P}_h(\mathbf{x}_t|\mathbf{x}_{t-h}) \, \mathrm{d}\mathbf{x}_{t-h} = \int p(\mathbf{x}_t, t)\mathbb{P}_h(\mathbf{x}_{t+h}|\mathbf{x}_t) \, \mathrm{d}\mathbf{x}_{t+h} = p(\mathbf{x}_t, t), \tag{15}$$

*can be interpreted as the GFlowNets flow matching constraint $\sum_{\mathbf{s}} F(\mathbf{s}, \mathbf{s}') = \sum_{\mathbf{s}''} F(\mathbf{s}', \mathbf{s}'') \triangleq F(\mathbf{s}'), \; \forall \mathbf{s}' \in \mathcal{S}$, where we have $F(\mathbf{s}, \mathbf{s}') = F(\mathbf{s})P_F(\mathbf{s}'|\mathbf{s})$.*

We point out that Eq. 15 is a standard starting point for deriving the Fokker-Planck equation (a similar discussion can be found in Karras et al. (2022)), as shown in Appendix:

**Proposition 8** (Øksendal (1985)). *Taking the limit as $h \to 0$, Eq. 15 implies*

$$\partial_t p(\mathbf{x}, t) = -\nabla_{\mathbf{x}} \left(p(\mathbf{x}, t)\mathbf{f}(\mathbf{x}, t)\right) + \frac{1}{2} \nabla_{\mathbf{x}}^2 \left(p(\mathbf{x}, t)g^2(\mathbf{x}, t)\right). \tag{16}$$

**Equivalence between detailed balance and score matching.** We investigate such a setting where we want to model the reverse process: $\mathrm{d}\mathbf{x} = \left[\mathbf{f}(\mathbf{x}) - g^2(t)\mathbf{s}(\mathbf{x}, t)\right] \mathrm{d}\bar{t} + g(t) \, \mathrm{d}\bar{\mathbf{w}}_t$, where $\mathbf{s}(\cdot, \cdot) : \mathbb{R}^D \times [0, 1] \to \mathbb{R}^D$ is a neural network, and $\mathbf{f}(\cdot)$ and $g(\cdot)$ are given[3]. We propose to use detailed balance to learn this neural network. From the above discussion, we can see there is an analogy realized by $F(\mathbf{s}_t) \approx p_t(\mathbf{x})$. We show the validity of such a strategy in the following proposition.

**Proposition 9.** *GFlowNets' detailed balance condition*

$$\lim_{h \to 0} \frac{1}{\sqrt{h}} \left(\log p_t(\mathbf{x}_t) + \log P_F(\mathbf{x}_{t+h}|\mathbf{x}_t) - \log p_{t+h}(\mathbf{x}_{t+h}) + \log P_B(\mathbf{x}_t|\mathbf{x}_{t+h})\right) = 0, \tag{17}$$

*($\forall \mathbf{x}_t \in \mathbb{R}^D, \forall t \in (0, 1)$) is equivalent to $\boldsymbol{\epsilon}^\top \left(\mathbf{s}(\mathbf{x}_t, t) - \nabla_{\mathbf{x}} \log p_t(\mathbf{x})\right) = 0, \forall \boldsymbol{\epsilon}, \mathbf{x}_t \in \mathbb{R}^D, \forall t \in [0, 1]$, which is the optimal solution to (sliced) score matching:*

$$\min_{\mathbf{s}} \mathbb{E}_{\mathbf{x} \sim p_t} \mathbb{E}_{\boldsymbol{\epsilon}} \left[\boldsymbol{\epsilon}^\top \nabla_{\mathbf{x}} \mathbf{s}(\mathbf{x})\boldsymbol{\epsilon} + \frac{1}{2} \left(\boldsymbol{\epsilon}^\top \mathbf{s}(\mathbf{x}, t)\right)^2\right], \forall t.$$

This result states that the equivalence between the optimal solutions of GFlowNet detailed balance and score matching. Notice that this is irrelevant to the specific optimization process.

---

[2]This is equivalent to defining $\tilde{\mathbf{x}}_t = (\mathbf{x}_t, t)$ and conduct discussion with $\tilde{\mathbf{x}}_t$ instead.

[3]$\mathbf{f}(\mathbf{x}), \mathbf{s}(\mathbf{x})$ could also be written as $\mathbf{f}(\mathbf{x}_t, t), \mathbf{s}(\mathbf{x}_t, t)$ in a more strict / general way.

### 3.2.3 Schrödinger Bridge

The Schrödinger Bridge or SB for short (Schrödinger, 1932; Léonard, 2013; Chen et al., 2021) is a classical problem which solves the entropy regularized optimal transport. In order to achieve this bridging target, the Iterative Proportional Fitting or IPF for short (Kullback, 1968) method proposes to solve the Schrödinger Bridge problem with the following alternating optimization:

$$\pi^{2m+1} = \arg\min\{\mathcal{D}_{\mathrm{KL}}\left(\pi\|\pi^{2m}\right) : \pi_{\mathrm{start}} = p_{\mathrm{prior}}\}, \tag{18}$$

$$\pi^{2m+2} = \arg\min\{\mathcal{D}_{\mathrm{KL}}\left(\pi\|\pi^{2m+1}\right) : \pi_{\mathrm{end}} = p^*\}, \tag{19}$$

where $\pi, \pi^m \in \mathcal{P}(\mathcal{X}^{n+1})$, namely distributions on $\mathcal{X}^{n+1}$ space, $\pi^0$ is some given base measure, and $\pi_{\mathrm{start}}, \pi_{\mathrm{end}} \in \mathcal{P}(\mathcal{X})$ refer to the marginal distribution of $\pi$ on the first and last index respectively.

In practice, IPF models the joint distribution $\pi$ in Eq. 18 in the decomposition form of forward probability product $\pi_{\mathrm{start}}(\mathbf{s}_1)\prod_i p(\mathbf{s}_{i+1}|\mathbf{s}_i)$, as in Eq. 5 and 7, as the marginal distribution on the first state is fixed. On the contrary, $\pi$ in Eq. 19 is modelled with backward probability decomposition $\pi_{\mathrm{end}}(\mathbf{s}_n)\prod_i p(\mathbf{s}_i|\mathbf{s}_{i+1})$, as in Eq. 6 and 8. As a matter of fact, such a discrete-time SB formulation generalizes the DDPM by relaxing the constraint on its noise diffusion process.

**Observation 10.** *The discrete-time Schrödinger Bridge is a special case of GFlowNet. Compared to the DDPM formulation, SB does not use a fixed backward policy, but learns both the forward and backward policies together.*

We remark that such alternating optimization is in the same spirit as the wake-sleep algorithm (Hinton et al., 1995) for learning latent variable models, without requiring something like the REINFORCE gradient estimator. Regarding the continuous-time setup, a similar observation could be made that SB generalizes the ScoreSDE formulation by not using a fixed noising process. We refer to Bortoli et al. (2021); Shi et al. (2022) for practical guidance regarding learning diffusion SB for generative modeling.

## 3.3 Exact Likelihood Models

Autoregressive (AR) models can b e viewed as sampling $p(\mathbf{x}_{1:i+1}|\mathbf{x}_{1:i})$ sequentially one dimension $i$ at a time in order to generate the final vector $\mathbf{x}$. Zhang et al. (2022b) use an AR-like (with a learnable ordering) model to parametrize the GFlowNet. Indeed, we can define every (forward) action of the GFlowNet as specifying one more pixel on top of the current state, and the backward policy turns one pixel into an unspecified value (Zhang et al., 2022b). This makes AR models special cases of GFlowNet where the order in which the pixels are specified is fixed, making the GFlowNet DAG a tree (Bengio et al., 2021a).

**Observation 11.** *The standard autoregressive model is a special kind of GFlowNet where*

- $\mathbf{s}_i := \mathbf{x}_{1:i}$ *is the GFlowNet state;*
- $P_F(\mathbf{s}_{i+1}|\mathbf{s}_i) = p(\mathbf{x}_{1:i+1}|\mathbf{x}_{1:i})$ *is the forward policy;*
- $P_B(\mathbf{s}_i|\mathbf{s}_{i+1}) = \delta\{\mathbf{s}_i$ *comprises the first $i$ dimensions of $\mathbf{s}_{i+1}\}$,*

*where $\delta\{\cdot\}$ is the Dirac Delta distribution (indicator function) for continuous (discrete) variables.*

This modeling makes the latent graph of the GFlowNet to be a tree; alternatively, if we allow a learnable ordering as with Zhang et al. (2022b), the trajectories in latent space form a general DAG. This is related to non-autoregressive modeling methods in the NLP community (Gu et al., 2018). It is also noticed by Chang et al. (2023) that AR modeling shares a similar spirit with diffusion models that it sequentially constructs data, which is a core idea of GFlowNet—treating inference problems as sequential decision-making problems.

The normalizing flow or NF for short (Dinh et al., 2015) is another way to sequentially construct desired data. It first samples $\mathbf{z}_1$ from a base distribution (usually the standard Gaussian), and then applies a series of invertible transformations $\mathbf{z}_{i+1} = \mathbf{f}(\mathbf{z}_i)$ until one finally obtains $\mathbf{x} := \mathbf{z}_n$, where $n$ denotes the number of transformation layers.

**Observation 12.** *The NF is a special kind of GFlowNet with deterministic forward and backward policies (except the first transition step), and $\mathbf{s}_i = \mathbf{z}_i$ are GFlowNet states.*

With the GFlowNet implementation of NF, the base distribution is the first (and only stochastic) step, while the other steps are deterministic. When both $P_F(\mathbf{z}_{i+1}|\mathbf{z}_i)$ and $P_B(\mathbf{z}_i|\mathbf{z}_{i+1})$ are deterministic (each being a Dirac at the value of some function applied to the conditioning argument) and match each other, it must be that they correspond to invertible functions. If we think of the whole normalizing flow as a GFlowNet with a one-step transition, then we could still make an equivalence between the MLE objective of NFs and the KL-trajectory balance of GFlowNets. We next discuss maximum likelihood estimation (MLE) of AR and NF models.

**About MLE training.** AR models and NFs are usually trained with MLE. Although the likelihood of general GFlowNets is intractable, we lower bound it:

$$\log p_T(\mathbf{x}) = \log \int_{\mathbf{x} \in \tau} P_F(\tau)\, \mathrm{d}\tau = \log \mathbb{E}_{P_B(\tau|\mathbf{x})}\left[\frac{P_F(\tau)}{P_B(\tau|\mathbf{x})}\right] \tag{20}$$

$$\geq \mathbb{E}_{P_B(\tau|\mathbf{x})}\left[\log \frac{P_F(\tau)}{P_B(\tau|\mathbf{x})}\right] = -\mathcal{D}_{\mathrm{KL}}\left(P_B(\tau|\mathbf{x})\|P_F(\tau)\right), \tag{21}$$

This also corresponds to a kind of trajectory balance objective (KL-trajectory balance) as in Proposition 2. Notice this derivation is applicable to all GFlowNet specifications rather than just exact likelihood models. An IWAE-type bound (Burda et al., 2016) is also applicable. When we are given data $\mathbf{x}$, we can directly use $\log P_B(\tau|\mathbf{x}) - \log P_F(\tau)$ as a sample-based training loss for $\mathbf{x} \in \mathfrak{D}$, with $\tau \sim P_B(\tau|\mathbf{x})$, to maximize a variational lower bound on the log-likelihood. Furthermore, if the $P_B(\tau|\mathbf{x})$ is deterministic, this corresponds to a fixed ordering (a single trajectory) to construct $\mathbf{x}$, which is the AR interpretation of a GFlowNet, and minimizing $\log P_B(\tau|\mathbf{x}) - \log P_F(\tau)$ is the same thing as maximizing the likelihood $\log P_F(\tau)$ with $\tau$ corresponding to $\mathbf{x}$, i.e., the MLE loss of AR models.

Summarizing, both AR models and NFs are GFlowNets with a tree-structured latent state graph, making every terminating state reachable by only one trajectory.

### 3.4   Learning a Reward Function from Data

Sampling from an energy-based model (EBM) can be obtained from a GFlowNet whose negative log of the reward function is the energy function. We could use any GFlowNet model including those discussed in previous sections, and jointly train it together with the EBM. For instance, in the EB-GFN (Zhang et al., 2022b) algorithm a GFlowNet is used to amortize the computational MCMC process of the EBM contrastive divergence training. The two models (EBM and GFlowNet) are updated alternately.

GANs (Goodfellow et al., 2014) are closely related to EBMs (Che et al., 2020), while its algorithm is more computationally efficient. However, though it may look reasonable at first glance, we cannot directly use the discriminator $D(\mathbf{x})$ as the reward for GFlowNet training. If we did, at the end of perfect training, we would get an optimal discriminator $D^*(\mathbf{x}) = \frac{p^*(\mathbf{x})}{p^*(\mathbf{x}) + p_T(\mathbf{x})}$, and the optimized GFlowNet terminating distribution would be $p_T(\mathbf{x}) \propto D^*(\mathbf{x})$. This cannot induce $p_T(\mathbf{x}) = p^*(\mathbf{x})$. In fact, if $p_T(\mathbf{x}) = p^*(\mathbf{x})$, we will have $D^*(\mathbf{x}) \equiv 1/2$ and $p_T(\mathbf{x}) = p^*(\mathbf{x}) \equiv$ constant, which is impossible for general data with unbounded support. We could instead conceive the following algorithm.

**Proposition 13.** *An alternative algorithm which trains the discriminator to distinguish between generated data and true data as in GANs, and trains a GFlowNet generator with negative energy of $\log \frac{D(\mathbf{x})}{1-D(\mathbf{x})} + \log p^*(\mathbf{x})$ would result in a valid generative model.*

Nonetheless, we unfortunately do not have access to the exact value of $p_T(\mathbf{x})$ if the generator is a general GFlowNet due to the intractability of the integration in Eq. 20. Such likelihood probability value can indeed be approximately estimated (see Section C.1); however, the additional estimator computation makes this algorithm described in the proposition to be very expensive, as every training step would require an extra estimation step. Therefore, our framework is not able to naturally contain all generative modeling methods such as GANs as explained in this section, which is a limitation.

---

**Algorithm 1** Generative modeling via maximum likelihood estimation based GFlowNet (MLE-GFN)

---

**Require:** Trainset $\mathfrak{D} = \{\mathbf{x}_i\}_i$, GFlowNet policy $P_F(\tau; \boldsymbol{\theta}), P_B(\tau|\mathbf{x}; \boldsymbol{\phi})$ with parameters $\boldsymbol{\theta}, \boldsymbol{\phi}$.
1: **repeat**
2:  Uniformly sample $\mathbf{x}$ from dataset $\mathfrak{D}$;
3:  $\triangle\boldsymbol{\theta} \leftarrow \nabla_{\boldsymbol{\theta}}\mathcal{D}_{\mathrm{KL}}\left(P_B(\cdot|\mathbf{x}; \boldsymbol{\phi})\|P_F(\cdot; \boldsymbol{\theta})\right)$;
4:  Sample backward trajectories $\tau, \tau' \sim P_B(\cdot|\mathbf{x}; \boldsymbol{\phi})$;
5:  $\triangle\boldsymbol{\phi} \leftarrow \nabla_{\boldsymbol{\phi}}\mathcal{L}_{\mathrm{TBC}}(\tau, \tau')$;
6:  Update $\boldsymbol{\theta}, \boldsymbol{\phi}$ with some optimizer;
7: **until** some convergence condition

---

Table 1: Results with seven 2D synthetic problems. We display the MMD (in units of $1 \times 10^{-4}$).

| Method | 2spirals | 8gaussians | circles | moons | pinwheel | swissroll | checkerboard |
|---|---|---|---|---|---|---|---|
| PCD | 2.160 | 0.954 | 0.188 | 0.962 | 0.505 | 1.382 | 2.831 |
| ALOE | 21.926 | 107.320 | 0.497 | 26.894 | 39.091 | 0.471 | 61.562 |
| ALOE+ | **0.149** | 0.078 | 0.636 | 0.516 | 1.746 | 0.718 | 12.138 |
| EB-GFN | 0.583 | 0.531 | 0.305 | 0.121 | 0.492 | 0.274 | 1.206 |
| MLE-GFN | 0.472 | **0.046** | **-0.026** | **-0.021** | **0.211** | **0.151** | **0.393** |

## 4 Towards Improved Generative Modeling with Insights from GFlowNets

We have provided a GFlowNet-based probabilistic framework to unify the generative behaviors of different classes of generative models. In this section, we investigate the possibility of further boosting the performance of existing generative modeling methods with insights from GFlowNets.

### 4.1 Trajectory Balance Consistency

Based on the difference between generative modeling with GFlowNet sampling discussed in previous text, it is hard to find a straightforward way to combine algorithms from these two worlds. To see this, recall that in the original GFlowNet trajectory balance objective, the reward value $R(\mathbf{x})$ is needed: $\mathcal{L}_{\mathrm{TB}}(\tau) = \left[\log \frac{ZP_F(\tau)}{R(\mathbf{x})P_B(\tau|\mathbf{x})}\right]^2$, where $Z$ is a learnable scalar parameter, $\tau = (\mathbf{s}_1, \ldots, \mathbf{s}_n)$ and $\mathbf{s}_n = \mathbf{x}$. However, this is not practicable since in generative modeling we do not have access to the callable target density function $p^*(\mathbf{x}) \propto R(\mathbf{x})$. To circumvent this obstacle, the energy-based GFlowNet (Zhang et al., 2022b, EB-GFN) additionally parameterize an energy-based model to approximate (the negative logarithm of) the reward function, and jointly train the GFlowNet as well as the energy function in an alternative way.

In this work, instead of explicitly learning an extra neural network to serve as the reward proxy, we notice that if a GFlowNet with infinite capacity is trained to completion, we would have $\frac{P_F(\tau)}{P_B(\tau|\mathbf{x})} = \frac{R(\mathbf{x})}{Z} = \frac{P_F(\tau')}{P_B(\tau'|\mathbf{x})}$ for any two different trajectories $\tau, \tau'$ with the same terminating state $\mathbf{x}$. Consequently, we propose such consistency objective to avoid the appearance of the reward term,

$$\mathcal{L}_{\mathrm{TBC}}(\tau, \tau') = \left[\log \frac{P_F(\tau)}{P_B(\tau|\mathbf{x})} - \log \frac{P_F(\tau')}{P_B(\tau'|\mathbf{x})}\right]^2, \tag{22}$$

where we use "TBC" to denote "trajectory balance consistency".

The proposed consistency loss objective only assures the balance between the forward and backward trajectories of GFlowNet model, but receives no signal about information of the target distribution that the GFlowNet desire to match. Hence we cannot use $\mathcal{L}_{\mathrm{TBC}}$ as the only training loss even with $\mathbf{x}$ taken from the training set, which would potentially end up with a naive solution, *e.g.*, constant output GFlowNet policy. Instead, we propose to combine the trajectory balance consistency (which trains the backward policy) with the original generative modeling methods (which train the forward policy). The TBC objective computation needs two trajectories and thus doubles the memory occupation, but it can be parallel sampled efficiently and thus no additional speed overhead. We will demonstrate the efficacy of the proposed strategy in the following section.

Table 2: Experiment results with seven 2D synthetic problems. We display the negative loglikelihood (NLL).

| Metric | Method | 2spirals | 8gaussians | circles | moons | pinwheel | swissroll | checkerboard |
|--------|--------|----------|-----------|---------|--------|----------|-----------|--------------|
| NLL↓ | PCD | 20.094 | 19.991 | 20.565 | 19.763 | 19.593 | 20.172 | 21.214 |
| | ALOE | 20.295 | 20.350 | 20.565 | 19.287 | 19.821 | 20.160 | 54.653 |
| | ALOE+ | 20.062 | 19.984 | 20.570 | 19.743 | 19.576 | 20.170 | 21.142 |
| | EB-GFN | **20.050** | 19.982 | **20.546** | 19.732 | **19.554** | **20.146** | 20.696 |
| | MLE-GFN | **20.050** | **19.965** | 20.554 | **19.719** | 19.555 | **20.144** | **20.682** |

Following the analysis from Malkin et al. (2023), we show the relationship between the proposed GFlowNet balance objective and divergence minimizing objective.

**Proposition 14.** *Denote the parameters of the backward policy $P_B$ by $\phi$, then the gradient of the $\mathcal{L}_{TBC}$ objective defined in Eq. 22 with respect to $\phi$ satisfies*

$$\frac{1}{4}\mathbb{E}_{\tau,\tau'\sim P_B}\left[\nabla_\phi \mathcal{L}_{TBC}(\tau,\tau')\right] = \nabla_\phi \mathcal{D}_{\mathrm{KL}}\left(P_B \| P_F\right). \tag{23}$$

The proposition demonstrates the validity of using $\mathcal{L}_{\mathrm{TBC}}$ to update the model. This indicates that, as a general rule of generative modeling, we could optimize the forward policy with the variational bound in Eq. 21, and optimize the backward policy with $\mathcal{L}_{\mathrm{TBC}}$ defined in Eq. 22. We specify our method in Algorithm 1. We refer to the algorithm as MLE-GFN, where MLE stands for maximum likelihood estimation, since it is essentially (1) optimizing a variational bound of the model likelihood on given data; (2) maximizing the model likelihood in a trajectory level, using backward trajectories generated from true data.

## 4.2 Synthetic Demonstration

The experiment in this subsection follows the setup from Dai et al. (2020); Zhang et al. (2022b). The objective is to model 7 different distributions over 32-dimensional binary space, as displayed in Figure 1. The binary data is quantized from 2 dimensional continuous data via the Gray code (Gray, 1953). We consider with the baselines from Zhang et al. (2022b), including persistent contrastive divergence or PCD for short (Tieleman, 2008), ALOE (Dai et al., 2020), and energy-based GFlowNet (EB-GFN).

For the proposed trajectory balance consistency augmented GFlowNet (MLE-GFN) method, we follow the same GFlowNet modeling as in EB-GFN, just using the novel objectives in Algorithm 1 to separately learn the forward and backward policies. Notice that EB-GFN needs to learn an additional energy function, thus consumes larger number of parameters. In the result presentation, "ALOE" and "ALOE+" denote two different modeling methods, where the former shares a similar number of parameters to the GFlowNets, while the latter is thirty times larger. For evaluation, we report the MMD (Gretton et al., 2012) between ground truth samples and generated samples. We demonstrate quantitative results in Table 1, where the proposed algorithm outperform most other baselines. For completeness, we also experiment on the larger dynamical MNIST binary probabilistic modeling benchmark, where our proposed MLE-GFN achieves an NLL of 100.23 compared to the original EB-GFN with an NLL of 105.75. We visualize the ground truth samples and GFlowNet generated samples in its 2-D form in Figure 1. We also demonstrate the likelihood evaluation results in Table 2, which indicates that the proposed method achieves a fairly good level of distribution fitting. Regarding training details, we use the Adam optimizer with $1 \times 10^{-4}$ and $1 \times 10^{-5}$ for learning the forward and backward policy, respectively. The training keeps $10^5$ steps. All other setups follow Zhang et al. (2022b).

## 4.3 DDPM Demonstration

DDPM (Ho et al., 2020) defines a diffusion generative process and has achieved great success in high quality visual generation. In Section 3.2, we have pointed out their similarity on modeling behavior: when we parameterize the forward and backward policy with particular Gaussian distributions as in Observation 3, a GFlowNet is crystallized into a DDPM. The forward policy $P_F$ is the denoised reverse process $p$, while the backward policy $P_B$ is the diffusion process $q$ which involves noises.

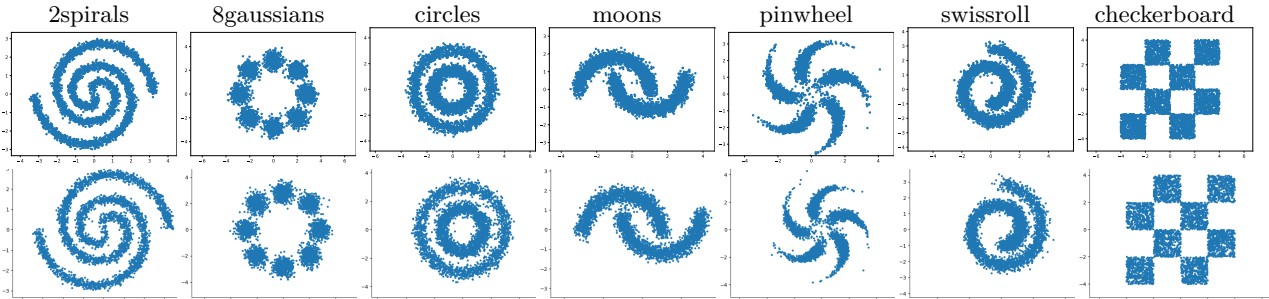

Figure 1: *Top:* Visualization of the samples for synthetic problems from ground truth. *Bottom:* Visualization of the samples generated with the proposed GFlowNet-based algorithm.

Table 3: Comparison between MLE-GFN in Algorithm 1 and baselines on CIFAR-10. We evaluate both the sample quality (FID) and likelihood (NLL). We train models with a smaller level of computation, which explains the performance gap with the results in the original DDPM paper.

| Method | FID↓ | NLL↓ |
|---|---|---|
| Baseline | 10.616 | 4.276 |
| iDDPM | 10.823 | 4.198 |
| MLE-GFN | **10.062** | **4.157** |

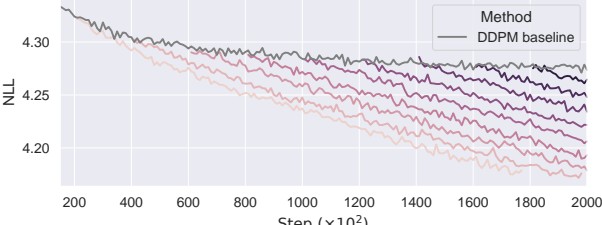

Figure 2: NLL of baseline and proposed methods. The gray line denotes the DDPM baseline; the colored lines denote the proposed GFlowNet consistency augmented training started with the pretrained weights of the baseline method from $20000, 40000, \ldots, 180000$ step, respectively.

DDPM use a fixed $P_B$ (and hence fixed $\{\beta_i\}_i$) to enable stable training[4]. In this section, we propose to utilize the proposed consistency objective to "defreeze" the constant for better expressiveness and sample quality, following the guideline in Algorithm 1. Concretely, under the context of DDPM modeling, the likelihood variational lower bound reduces to the original DDPM denoising objective (see Proposition 4), and the consistency objective is used to update the variance parameter $\{\beta_i\}_i$. Notice that there are other works, *e.g.*, Nichol & Dhariwal (2021), that also propose to achieve a similar goal, but from different angles. In this section, we are not aiming for a state-of-the-art performance, but mainly for a demonstration of the application of GFlowNet idea upon generative modeling tasks.

Due to computation resource limitation, we train a DDPM-specified GFlowNet on the CIFAR-10 dataset with $n = 100$ (*i.e.*, GFlowNet trajectory length) denoising steps for 200k training steps. This is less than the $n = 1000$ and 800k step training by the original diffusion. We defer other training and experimental details to Section C. We first directly quantitatively show the sample quality and likelihood evaluation metric of the proposed GFlowNet-inspired consistency augmented algorithm and the DDPM baseline in Table 3. Here the sample quality is measured by the FID score (Heusel et al., 2017). Apart from the DDPM baseline, we also compare with the hybrid objective proposed in improved DDPM (Nichol & Dhariwal, 2021, iDDPM), which is designed to improve the likelihood evaluation metric. iDDPM will actually achieve worse sample quality while having better likelihood; on the other hand, our proposed approach could achieve a better trade-off, achieving both better likelihood and FID performance.

What's more, we also try to finetune a pretrained model with our proposed method. Specifically, we train a DDPM model with a fixed number of steps, and then switch to the proposed MLE-GFN to continue the training. Figure 2 presents the curves of likelihood evaluation for the DDPM baseline (grey curve) and the proposed method (violet curves).

---

[4]It is mentioned in Section 4.2 of Ho et al. (2020) that "incorporating a parameterized variance leads to unstable training and poor sample quality".

Table 4: ImageNet-32 results.

| Method | FID↓ | NLL↓ |
|---|---|---|
| Baseline | 17.65 | 4.57 |
| iDDPM | 17.68 | 4.50 |
| MLE-GFN | 16.36 | 4.47 |

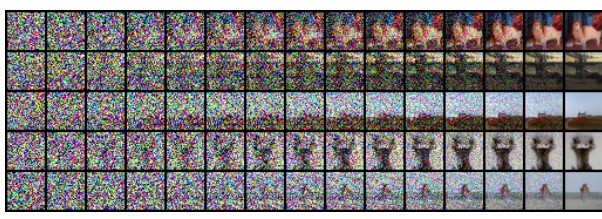 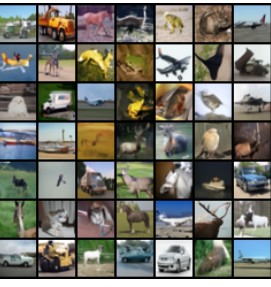 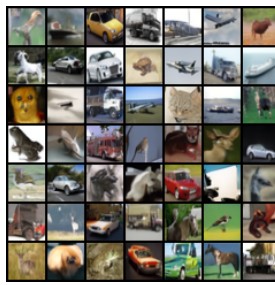

Figure 3: *Left:* visualization of trajectory examples on CIFAR-10 image space. *Right:* MLE-GFN generated samples.

We train on a single V100 GPU for 200k steps, which takes less than three days. We use the Adam optimizer with a learning rate of $2 \times 10^{-4}$ for updating the forward policy and a learning rate of $2 \times 10^{-5}$ for updating the backward policy. All NLL results are computed in the unit of bits per dim (BPD). The denoising process (forward policy) is parameterized with a UNet as done by Ho et al. (2020). The backward policy is parameterized in the way that its parameter $\phi = \{\phi_i\}_i, \beta_i = \bar{\beta}_i \cdot \exp(\phi_i)$, where $\{\bar{\beta}_i\}_i$ are the original variance parameters from Ho et al. (2020). We also visualize several trajectories of the GFlowNet in Figure 3 (left), where the intermediate state is updated with the forward policy from the left-hand side to the right-hand side of the figure. In Figure 3 (right), we visualize several generated image samples from MLE-GFN training on CIFAR-10 dataset. To further show the scaling ability of the proposed method, we also conduct a similar experiment on ImageNet-32 dataset and show the results in Table 4, see details in Lahlou et al. (2023a).

## 5 Conclusion

We have interpreted existing generative models as GFlowNets with different policies over sample trajectories. This provides some insight into the overlaps between existing generative modeling frameworks, and their connection in the sense of training algorithms. Furthermore, this GFlowNet perspective inspires us to propose a tractable generative modeling algorithm, of which we corroborate the effectiveness on both discrete and continuous settings.

**Broader Impact Statement**

Most of this paper focuses on theoretical connection between the GFlowNet framework and existing generative models, thusits implications are largely confined to the realm of academic research and technical advancement. Consequently, the immediate societal impact of this aspect is expected to be neutral, as it is focused on advancing knowledge within the machine learning community without direct consequences on wider society. The second part of this paper is about how to improve generative modeling with insights from GFlowNets. Improved generative models have the potential to impact fields such as art, entertainment, and data synthesis, contributing to the creation of realistic images, audio, and data that could be harnessed for various purposes. Possible potential aspects include: fake content generation, which could lead to misuse; privacy concerns; loss of creative ownership of art works; job displacement of artists and writers. It's important to note that many of these potential negative effects are not inherent to generative models themselves, but rather arise from how they are used and misused. Responsible research, ethical considerations, and regulatory frameworks can play a crucial role in minimizing these negative impacts and ensuring the technology's safe and beneficial integration into society.

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

## A Summary of Notation

| Symbol | Description |
|---|---|
| $\mathcal{S}$ | GFlowNet state space |
| $\mathcal{X}$ | object (terminal state) space, subset of $\mathcal{S}$ |
| $\mathcal{A}$ | action / transition space (edges $\mathbf{s} \rightarrow \mathbf{s}'$) |
| $\mathcal{G}$ | directed acyclic graph $(\mathcal{S}, \mathcal{A})$ |
| $\mathcal{T}$ | set of complete trajectories |
| $\mathbf{s}$ | state in $\mathcal{S}$ |
| $\mathbf{s}_0$ | initial state, element of $\mathcal{S}$ |
| $\mathbf{x}$ | terminal state in $\mathcal{X}$ |
| $\tau$ | trajectory in $\mathcal{T}$ |
| $F : \mathcal{T} \rightarrow \mathbb{R}$ | Markovian flow |
| $F : \mathcal{S} \rightarrow \mathbb{R}$ | state flow |
| $F : \mathcal{A} \rightarrow \mathbb{R}$ | edge flow |
| $P_F(\mathbf{s}'|\mathbf{s})$ | forward policy (distribution over children) |
| $P_B(\mathbf{s}|\mathbf{s}')$ | backward policy (distribution over parents) |
| $P_T(\mathbf{x})$ | terminating distribution |
| $Z$ | scalar, equal to $\sum_{\tau \in \mathcal{T}} F(\tau)$ for a Markovian flow |

## B Proofs

### B.1 Proposition 2

*Proof.* We have

$$
\begin{aligned}
\mathcal{D}_{\mathrm{KL}}\left(P_B(\tau)\|P_F(\tau)\right) &= \mathbb{E}_{p^*(\mathbf{x})q(\mathbf{z}_{1:n-1}|\mathbf{x})}\left[\log \frac{p^*(\mathbf{x})q(\mathbf{z}_{1:n-1}|\mathbf{x})}{p(\mathbf{z}_{1:n-1})p(\mathbf{x}|\mathbf{z}_{1:n-1})}\right] \\
&= -\mathbb{E}_{p^*(\mathbf{x})}\mathbb{E}_{q(\mathbf{z}_{1:n-1}|\mathbf{x})}\left[\log \frac{p(\mathbf{x}, \mathbf{z}_{1:n-1})}{q(\mathbf{z}_{1:n-1}|\mathbf{x})}\right] + \mathbb{E}_{p^*(\mathbf{x})}\left[\log p^*(\mathbf{x})\right] \\
&= -\mathbb{E}_{p^*(\mathbf{x})}\left[\mathrm{ELBO}_{p,q}(\mathbf{x})\right] - \mathcal{H}[p^*(\cdot)].
\end{aligned}
$$

Here $\mathcal{H}[p^*(\cdot)]$ denotes the entropy of the target distribution, which is a constant w.r.t. GFlowNet parameters. □

### B.2 Proposition 4

We follow a similar derivation of Ho et al. (2020) and Section B.1.

$$
\mathcal{D}_{\mathrm{KL}}\left(P_B(\tau)\|P_F(\tau)\right) \cong -\mathbb{E}_{p^*(\mathbf{x})}\left[\mathrm{ELBO}_{p,q}(\mathbf{x})\right]
$$

$$
\begin{aligned}
\mathrm{ELBO}_{p,q}(\mathbf{x}) &= \mathbb{E}_q\left[-\log \frac{p(\mathbf{z}_{1:n})}{q(\mathbf{z}_{1:n-1}|\mathbf{x})}\right] = \mathbb{E}_q\left[-\log p(\mathbf{z}_1) - \sum_{i<L}\log \frac{p(\mathbf{z}_{i+1}|\mathbf{z}_i)}{q(\mathbf{z}_i|\mathbf{z}_{i+1})} - \log \frac{p(\mathbf{z}_n|\mathbf{z}_L)}{q(\mathbf{z}_L|\mathbf{z}_n)}\right] \\
&= \mathbb{E}_q\left[-\log \frac{p(\mathbf{z}_1)}{q(\mathbf{z}_1|\mathbf{z}_n)} - \sum_{i<L}\log \frac{p(\mathbf{z}_{i+1}|\mathbf{z}_i)}{q(\mathbf{z}_{i+1}|\mathbf{z}_i, \mathbf{z}_n)} - \log p(\mathbf{z}_n|\mathbf{z}_L)\right] \\
&\cong \mathbb{E}_q\left[\sum_{i<L}\mathcal{D}_{\mathrm{KL}}\left(q(\mathbf{z}_{i+1}|\mathbf{z}_i, \mathbf{z}_n)\|p(\mathbf{z}_{i+1}|\mathbf{z}_i)\right) - \log p(\mathbf{z}_n|\mathbf{z}_L)\right]
\end{aligned}
$$

Since the KL divergence between two Gaussian distributions has a close form, we have

$$\mathcal{D}_{\mathrm{KL}}\left(q(\mathbf{z}_{i+1}|\mathbf{z}_i,\mathbf{z}_n)\|p(\mathbf{z}_{i+1}|\mathbf{z}_i)\right) \cong C_i \cdot \mathbb{E}\left[\|\boldsymbol{\mu}_i(\mathbf{z}_i)-\tilde{\boldsymbol{\mu}}_i(\mathbf{z}_i)\|^2\right],$$

$$\mathbb{E}_q\left[-\log p(\mathbf{z}_n|\mathbf{z}_L)\right] \cong C_L \cdot \mathbb{E}\left[\|\boldsymbol{\mu}_L(\mathbf{z}_i)-\mathbf{z}_n\|^2\right],$$

where $\{C_i\}_i$ are scalar constants, and $\tilde{\boldsymbol{\mu}}_i(\mathbf{z}_i)$ is the mean value of $q(\mathbf{z}_{i+1}|\mathbf{z}_i,\mathbf{z}_n)$, and $\tilde{\boldsymbol{\mu}}_i(\mathbf{z})=\tilde{C}_i^1\mathbf{z}+\tilde{C}_i^2\mathbf{x}$ is a fixed (*i.e.*, non-learnable) transformation defined with $\{\beta_i\}_i$ and $\mathbf{x}=\mathbf{z}_n$. Here $\tilde{C}_i^1,\tilde{C}_i^2$ are deterministic functions of $\{\beta_i\}_i$ defined by Ho et al. (2020). This derivation indicates that training a GFlowNet with KL-trajectory balance objective would end up with an effectively same (*i.e.*, ignoring the constants) objective with the regression loss proposed in denoised autoencdoer (Vincent et al., 2008) and denoising diffusion probabilistic models.

## B.3 Proposition 8

*Proof.* First notice

$$\partial_t p(\mathbf{x},t) \triangleq \lim_{h\to 0}\frac{1}{h}\left(p(\mathbf{x},t+h)-p(\mathbf{x},t)\right) = \lim_{h\to 0}\frac{1}{h}\left(\int p(\mathbf{x}',t)\mathbb{P}_h(\mathbf{x}|\mathbf{x}')\,\mathrm{d}\mathbf{x}' - p(\mathbf{x},t)\right).$$

Then for any function $w(\mathbf{x})$, we have

$$\int w(\mathbf{x})\partial_t p(\mathbf{x},t)\,\mathrm{d}\mathbf{x}$$
$$= \int w(\mathbf{x})\lim_{h\to 0}\frac{1}{h}\left(\int p(\mathbf{x}',t)\mathbb{P}_h(\mathbf{x}|\mathbf{x}')\,\mathrm{d}\mathbf{x}' - p(\mathbf{x},t)\right)\mathrm{d}\mathbf{x}$$
$$= \lim_{h\to 0}\frac{1}{h}\left(\int w(\mathbf{x})\int p(\mathbf{x}',t)\mathbb{P}_h(\mathbf{x}|\mathbf{x}')\,\mathrm{d}\mathbf{x}'\,\mathrm{d}\mathbf{x} - \int w(\mathbf{x}')p(\mathbf{x}',t)\int \mathbb{P}_h(\mathbf{x}|\mathbf{x}')\,\mathrm{d}\mathbf{x}\,\mathrm{d}\mathbf{x}'\right)$$
$$= \lim_{h\to 0}\frac{1}{h}\int p(\mathbf{x}',t)\mathbb{P}_h(\mathbf{x}|\mathbf{x}')\left(w(\mathbf{x})-w(\mathbf{x}')\right)\mathrm{d}\mathbf{x}\,\mathrm{d}\mathbf{x}'$$
$$\triangleq \int p(\mathbf{x}',t)\sum_{n=1}w^{(n)}(\mathbf{x}')D_n(\mathbf{x}')\mathbf{x}'$$
$$= \int w(\mathbf{x}')\sum_{n=1}\left(-\frac{\partial}{\partial\mathbf{x}'}\right)^n\left(p(\mathbf{x}',t)D_n(\mathbf{x}')\right)\mathrm{d}\mathbf{x}',$$

where $D_n(\mathbf{x}')=\lim_{h\to 0}\frac{1}{hn!}\int\mathbb{P}_h(\mathbf{x}|\mathbf{x}')(\mathbf{x}-\mathbf{x}')^n\,\mathrm{d}\mathbf{x}$ and the last step uses integral by parts. This tells us

$$\partial_t p(\mathbf{x},t)=\sum_{n=1}\left(-\frac{\partial}{\partial\mathbf{x}}\right)^n\left(p(\mathbf{x},t)D_n(\mathbf{x})\right)=-\nabla_\mathbf{x}\left(p(\mathbf{x},t)\mathbf{f}(\mathbf{x},t)\right)+\frac{1}{2}\nabla_\mathbf{x}^2\left(p(\mathbf{x},t)g^2(\mathbf{x},t)\right),$$

which is essentially the Fokker-Planck equation.

$\square$

## B.4 Proposition 9

From the above SDEs, we know that the forward and backward policy is

$$\mathbf{x}_{t+h}=\mathbf{x}_t+\mathbf{f}(\mathbf{x}_t)h+\sqrt{h}g(t)\cdot\boldsymbol{\delta}_F,$$
$$\mathbf{x}_t=\mathbf{x}_{t+h}+\left[g^2(t+h)\mathbf{s}(\mathbf{x}_{t+h})-\mathbf{f}(\mathbf{x}_{t+h})\right]h+\sqrt{h}g(t+h)\boldsymbol{\delta}_B,$$

where $\boldsymbol{\delta}_F, \boldsymbol{\delta}_B \sim \mathcal{N}(0, \mathbf{I}_D)$. Since we know $h \to 0$, the left and right side of Eq. 17 become

$$\log P_F(\mathbf{x}_{t+h}|\mathbf{x}_t) - \log P_B(\mathbf{x}_t|\mathbf{x}_{t+h}) = -\frac{D}{2}\log(2\pi g_t^2 h) - \frac{1}{2g_t^2 h}\|\Delta\mathbf{x} - h\mathbf{f}_t\|^2$$
$$+ \frac{D}{2}\log(2\pi g_{t+h}^2 h) + \frac{1}{2g_{t+h}^2 h}\|\Delta\mathbf{x} - h\mathbf{f}_{t+h} + hg_{t+h}^2\mathbf{s}(\mathbf{x}_{t+h})\|^2,$$

$$\log p_{t+h}(\mathbf{x}_{t+h}) - \log p_t(\mathbf{x}_t) = \Delta\mathbf{x}^\top \nabla_{\mathbf{x}} \log p_t(\mathbf{x}) + O(h),$$

where $g_t = g(t), \mathbf{f}_t = \mathbf{f}(\mathbf{x}_t), \Delta\mathbf{x} = \mathbf{x}_{t+h} - \mathbf{x}_t = \sqrt{h}g_t\boldsymbol{\epsilon}, \boldsymbol{\epsilon} \sim \mathcal{N}(0, \mathbf{I}_D)$. Therefore,

$$\lim_{h\to 0} \frac{1}{\sqrt{h}g_t}(\log p_{t+h}(\mathbf{x}_{t+h})x - \log p_t(\mathbf{x}_t)) = \boldsymbol{\epsilon}^\top \nabla_{\mathbf{x}} \log p_t(\mathbf{x}),$$

$$\lim_{h\to 0} \frac{1}{\sqrt{h}g_t}(\log P_F(\mathbf{x}_{t+h}|\mathbf{x}_t) - \log P_B(\mathbf{x}_t|\mathbf{x}_{t+h}))$$
$$= \lim_{h\to 0} \frac{D}{\sqrt{h}g_t}(\log g_{t+h} - \log g_t) - \frac{1}{2g_t^3 h^{3/2}}\left(hg_t^2\|\boldsymbol{\epsilon}\|^2 + h^2\|\mathbf{f}_t\|^2 - 2h^{3/2}g_t\boldsymbol{\epsilon}^\top\mathbf{f}_t\right)$$
$$+ \frac{1}{2g_t g_{t+h}^2 h^{3/2}}\left(hg_t^2\|\boldsymbol{\epsilon}\|^2 - 2h^{3/2}g_t\boldsymbol{\epsilon}^\top\mathbf{f}_{t+h} + 2g_t g_{t+h}^2 h^{3/2}\boldsymbol{\epsilon}^\top\mathbf{s}(\mathbf{x}_{t+h})\right)$$
$$= \lim_{h\to 0} \boldsymbol{\epsilon}^\top\left(\frac{\mathbf{f}_t}{g_t^2} - \frac{\mathbf{f}_{t+h}}{g_{t+h}^2}\right) + \boldsymbol{\epsilon}^T\mathbf{s}(\mathbf{x}_{t+h}) = \boldsymbol{\epsilon}^T\mathbf{s}(\mathbf{x}_t),$$

with some smoothness assumptions on $g(t)$ and $\mathbf{f}(\mathbf{x}_t)/g(t)$. This tells us that in order to satisfy the detailed balance criterion, we need to satisfy

$$\boldsymbol{\epsilon}^\top(\mathbf{s}(\mathbf{x}_t, t) - \nabla_{\mathbf{x}} \log p_t(\mathbf{x})) = 0, \forall \boldsymbol{\epsilon}, \mathbf{x}_t \in \mathbb{R}^D, \forall t \in [0, 1].$$

Since $\boldsymbol{\epsilon}$ could take any value, this is equivalent that the model $\mathbf{s}$ should match with score function $\nabla_{\mathbf{x}} \log p(\mathbf{x})$. Also, note that this is exactly sliced score matching (Song et al., 2019), which has a more practical formulation

$$\mathbb{E}_{\boldsymbol{\epsilon}}\mathbb{E}_{\mathbf{x}\sim p}\left[\boldsymbol{\epsilon}^\top \nabla_{\mathbf{x}}\mathbf{s}(\mathbf{x})\boldsymbol{\epsilon} + \frac{1}{2}\left(\boldsymbol{\epsilon}^\top\mathbf{s}(\mathbf{x})\right)^2\right].$$

### B.5  Proposition 13

When both models (the GFlowNet and the discriminator) are trained perfectly, we have

$$D^*(\mathbf{x}) = \frac{p^*(\mathbf{x})}{p^*(\mathbf{x}) + p_T(\mathbf{x})},$$

and thus

$$p_T(\mathbf{x}) \propto \exp\left(\log\frac{D^*(\mathbf{x})}{1 - D^*(\mathbf{x})} + \log p_T(\mathbf{x})\right) = \exp\left(\log\frac{p^*(\mathbf{x})}{p_T(\mathbf{x})} + \log p_T(\mathbf{x})\right) = p^*(\mathbf{x}).$$

Hence it is a valid generative model algorithm.

### B.6  Proposition 14

Because

$$\nabla_\phi \mathcal{L}_{\mathrm{TBC}}(\tau, \tau') = 2\left(\log\frac{P_F(\tau)}{P_B(\tau|\mathbf{x};\phi)} - \log\frac{P_F(\tau')}{P_B(\tau'|\mathbf{x};\phi)}\right)(-\nabla_\phi \log P_B(\tau|\mathbf{x};\phi) + \nabla_\phi \log P_B(\tau'|\mathbf{x};\phi)),$$

we have

$$\frac{1}{2}\mathbb{E}_{\tau,\tau'\sim P_B}\left[\nabla_\phi \mathcal{L}_{\text{TBC}}(\tau,\tau')\right]$$

$$= -\mathbb{E}_{\tau\sim P_B}\left[\log\frac{P_F(\tau)}{P_B(\tau|\mathbf{x};\phi)}\nabla_\phi\log P_B(\tau|\mathbf{x};\phi)\right] + \mathbb{E}_{\tau,\tau'\sim P_B}\left[\log\frac{P_F(\tau)}{P_B(\tau|\mathbf{x};\phi)}\nabla_\phi\log P_B(\tau'|\mathbf{x};\phi)\right]$$

$$\quad - \mathbb{E}_{\tau'\sim P_B}\left[\log\frac{P_F(\tau')}{P_B(\tau'|\mathbf{x};\phi)}\nabla_\phi\log P_B(\tau'|\mathbf{x};\phi)\right] + \mathbb{E}_{\tau,\tau'\sim P_B}\left[\log\frac{P_F(\tau')}{P_B(\tau'|\mathbf{x};\phi)}\nabla_\phi\log P_B(\tau|\mathbf{x};\phi)\right]$$

$$= -2\mathbb{E}_{\tau\sim P_B}\left[\log\frac{P_F(\tau)}{P_B(\tau|\mathbf{x};\phi)}\nabla_\phi\log P_B(\tau|\mathbf{x};\phi)\right] + \mathbb{E}_{\tau\sim P_B}\left[\log\frac{P_F(\tau)}{P_B(\tau|\mathbf{x};\phi)}\right]\underbrace{\mathbb{E}_{\tau'\sim P_B}\left[\nabla_\phi\log P_B(\tau'|\mathbf{x};\phi)\right]}_{=0}$$

$$= 2\nabla_\phi\mathbb{E}_{\tau\sim P_B}\left[\log\frac{P_B(\tau|\mathbf{x};\phi)}{P_F(\tau)}\right]$$

$$= 2\nabla_\phi\mathcal{D}_{\text{KL}}\left(P_B\|P_F\right)$$

## C  More about Experimental Demonstration

### C.1  Likelihood Estimation

In order to estimate the GFlowNet likelihood of given data $\mathbf{x}$, notice that the terminating probability (Eq. 20) can be estimated via importance sampling:

$$P_T(\mathbf{x}) = \mathbb{E}_{P_B(\tau|\mathbf{x})}\frac{P_F(\tau)}{P_B(\tau\mid\mathbf{x})} \approx \frac{1}{M}\sum_{j=1}^{M}\frac{P_F(\tau^j)}{P_B(\tau^j\mid\mathbf{x})}, \tag{24}$$

where $\tau^j \sim P_B(\tau\mid\mathbf{x})$ and $M$ is the number of particles (trajectories) used for the estimation. Such estimator is a valid lower bound of the true likelihood, and larger $M$ leads to a tighter estimation (Burda et al., 2016). In this work, we set $M = 100$ which is large enough for convergence of the estimation; for example on 8gaussians, the NLL estimation is 19.965798 when $M = 20$, is 10.965463 when $M = 100$, is 19.965334 when $M = 500$.

### C.2  Synthetic Demonstration

One may argue that it is hard to see the advantage of MLE-GFN over EB-GFN; however, the likelihood results in synthetic tasks are pretty close to "saturation", that is, different methods lead to similar NLL that are near the theoretical minimum despite different qualitative performance. The improvement in NLL is on a smaller scale due to the nature of the calculation. In this case, it is not meaningful anymore to focus on NLL, and we should resort to more sensitive metrics such as MMD. MMD and NLL capture different aspects of the generative model's performance. MMD measures the discrepancy between the distribution of generated samples and ground truth samples, focusing on pairwise sample similarities. NLL, on the other hand, measures the probability assigned to the ground truth samples by the generative model. The MMD is a more global metric, comparing the overall distributions of the generated and ground truth samples. NLL, however, is a local metric that can be sensitive to small changes in the model's output probabilities. It's probable that the proposed method has achieved global improvements in the generated samples' distribution, reflected by the better MMD scores. The proposed method reduces mode collapse, which leads to better MMD scores. However, the NLL might not improve significantly because the overall likelihood of the ground truth samples hasn't changed much. It's important to note that a model with reduced mode collapse may be considered more desirable in practice despite little improvement in NLL.

