# OpenReview forum: "Unifying Generative Models with GFlowNets and Beyond"
_TMLR — Rejected by TMLR_

### Review · Reviewer_mPir · 2023-09-16

**Summary Of Contributions:**

This paper aims at generalizing several types of generative models -- VAEs, diffusion, autoregressive and normalizing flows -- in a common framework based on GFlowNets (Bengio et al., 2021a). This equivalence relies on highlighting in each model an forward and a backward autoregressive sampling paths that are naturally expressed in the graph-based sampling model of GFlowNets. In some cases, the authors go further by proving a loss equivalence between the generative model and the associated GFlowNet.

Unfortunately, this generalization does not allow to learn a generative model with GFlowNet because the latter relies on the unnormalized, and unknown, data density function. Nevertheless, a GFlowNet-inspired regularization of standard generative models is introduced to support the relevance of this new perspective. It is shown to improve and outperform established baselines on toy and real-world generative modeling tasks.

**Audience:**

Yes

**Claims And Evidence:**

No

**Requested Changes:**

Please refer to the above weaknesses for details. To summarize, in decreasing order of importance:
- the central claims of the paper should imperatively be clarified or adjusted since they are weakly supported;
- it is important to improve the exposition of GFlowNets for clarity purposes;
- some results should be reformulated or better specified, but this should be easily fixable;
- I would appreciate an answer on the other issues I pointed out in the authors' response and in the paper.

**Strengths And Weaknesses:**

**Foreword.** My expertise is deep generative modeling, and I am not familiar with GFlowNets; I apologize in advance for any misunderstanding and look forward to discussing with the authors and other reviewers. I thoroughly studied the main paper but could only skim through the appendix.

## Strengths

To the best of my understanding and except for particular cases detailed below, the introduced theoretical results are **mainly sound and clearly presented**. They highlight the closeness between GFlowNets and several generative models, which might be **of interest in one community or another**.

The authors demonstrate the **usefulness of their GFlowNet-inspired regularization**, including results on CIFAR10 and a small ImageNet dataset. This is thanks to the noise levels throughout the diffusion process that could be learned with this regularization. As such, this idea has the **potential to drive further research and improvement on diffusion models**.

## Weaknesses

Unfortunately, this paper suffers from important weaknesses, listed below, which would require significant changes. Most importantly, the main claim of the paper -- that this GFlowNet perspective can provide new insights about the listed generative models -- is weakly supported by both the theoretical and empirical results.

### Concerns on the Main Claim - Lack of New Insights on Deep Generative Models

Let me first recall the following main claims in the paper.

> **(1.)** This analysis sheds light on their overlapping traits and provides a unifying viewpoint [...]. (abstract)

> **(2.)** [...] we show how many existing generative models could be taken as special cases of GFlowNets [...]. (p. 2)

> **(3.)** [...] this unification implies a method of constructing an agglomeration of varying types of generative modeling approaches [...].

To the best of my understanding, these claims are weakly supported by the paper material, as explained in the following.

**(1.)** While the shown connections are probably valid, this perspective does not bring new insights on the connection between all considered generative models. Indeed, these connections are already known and widely accepted in the literature (as hinted in the submission).
- The similarity between hierarchical VAEs and denoising diffusion models already appeared through the ELBO construction and variational inference in the original paper of Ho et al. (2020). It has then been pursued in many following works by many following works (Huang et al., 2021; Kingma et al., 2021; Song et al., 2021b). In this regard, the results of the paper on this connection may overlap with the prior work of Malkin et al. (2023) and Zimmermann et al. (2022), which would deserve further discussion.
- The connection between score-based diffusion models and denoising diffusion models was already noted by Song et al. (2020). This was also noted, among others, in the more recent work of Karras et al. (2022).
- The fact that Schrödinger Bridges generalize diffusion models was noted in the original paper of De Bortoli et al. (2021).
- The (light) similarity between diffusion and autoregressive models -- which both progressively construct a synthetic datapoint -- was already noticed by e.g. Chang et al. (2023).

The results presented in the paper do not provide any further insight on this matter.

**(2.)** The strength of the generalization results of GFlowNets depends on the considered generative model. For some of them (hierarchical VAEs and diffusion models), the authors show a model and loss equivalence. For others (Schrödinger Bridges and exact likelihood models), this equivalence only holds for the model.

Equivalences in terms of model (e.g. Observation 1 or 11) are actually straightforward. Indeed, GFlowNets are standard forward and backward graphical models and hence naturally embed (latent) autoregressive generation in models like diffusion, VAEs and normalizing flows. This provides little information as no new perspective in provided either on the GFlowNet or generative modeling side.

The originality of GFlowNets lies in their learning of reward-based sampling in such graphical models. As such, loss equivalence results between HVAEs / diffusion and GFlowNets are more meaningful. However, they remain limited as they solely rely on matching forward and backward probability paths, which is already a known core component of the studied models and is deeply rooted in their hierarchical structure.

**(3.)** Accordingly, the GFlowNet motivation for MLE-GFN is weak. The introduced TBC constraint of Eq. (21) relies on the observation of an equality between probability ratio in a perfectly trained GFlowNet. However, this constraint is not specific to GFlowNet: it should be satisfied by any perfect diffusion model. Therefore, while MLE-GFN might be valuable on its own, its link with GFlowNet remains limited, similar to the previously developed theory.

Song et al. (2021b) Denoising diffusion implicit models. ICLR 2021.\
Karras et al. Elucidating the Design Space of Diffusion-Based Generative Models. NeurIPS 2022.\
Chang et al. Muse: Text-to-image generation via masked generative transformers. arXiv, 2023.\
Zimmermann et al. A variational perspective on generative flow networks. TMLR, 2023.

### Lacking Background Clarity

While the paper is mostly clear from Section 3 onward, I found the introduction of GFlowNets hard to follow, without sufficient context. In the current state of the paper, it will be difficult for readers that are only familiar with deep generative models to obtain the required understanding of GFlowNets for the paper. A more detailed and accessible summary of GFlowNets, even in the appendix, should improve the readability of the paper.

### Concerns on Results Formulation

The formulation of some results might be confusing or lacking details, as explained below.
1. As rightly pointed out after Observation 6, the state of the graph must be augmented with time to impose make it a DAG. However, this constraint should probably be added to all other considered generative models except the AR one: given that they operate on continuous spaces, there is no guarantee that the graph is a DAG.
2. It should be specified between Observation 7 and Proposition 8 that the Fokker-Planck equation is already known to describe the probability paths of diffusion models (like in Karras et al., 2023).
3. Proposition 9 should be reformulated. To my understanding, this result simply implies that the score network perfectly matches the score (as expected when the forward and backward probability paths match). The equivalence is not about the optimization problem of sliced score matching (which is in any case irrelevant since such diffusion models use denoising score matching), but about score matching at optimality with infinite capacity.
4. The paragraph on MLE training on page 7 actually corresponds to an ELBO.

Karras et al. Elucidating the Design Space of Diffusion-Based Generative Models. NeurIPS 2022.

### Other Concerns

The following is a list of other minor issues in the paper, in no particular order.

1. While the introduction of GFlowNets suggests that it operates on discrete spaces, the authors prove equivalences for continuous spaces throughout the paper. While I see no reason why this should not work, could the authors confirm in the paper that GFlowNets are properly operating on continuous spaces?
2. I do not understand the sentence "It is well known that this hierarchical ELBO can also be represented as [some KL] [...]." To my knowledge, this is not the ELBO and I do not understand its role in the reasoning.
3. In the paragraph on discrete-space diffusion, it is stated that "$s_i$ denotes the one-hot representation of $s_i$". There seems to be a typo.
4. Could the authors explain why $\theta$ does not receive gradients from the TBC loss, and conversely why $\phi$ does not receive gradients from the KL? Unfolding the VAE analogy, it does not seem consistent as in VAEs both sets of parameters are jointly optimized.
5. Why are some MMD values in Table 1 negative? And what is the chosen kernel for the MMD?
6. Section 5.3 refers to "Table 5.3" in the third paragraph, but Table 5.3 does not exist.

---

> ### Comment · Reviewer_mPir · 2023-11-07
> **Comment on Revision**
>
> I would like to thank the authors for their answer and revision, which I carefully read. As things stand, while the updated submission showed slight improvements, they are unfortunately superficial. Given my review and other reviewer's comments, I think that the paper needs a significant overhaul to make its claims clear and accurately supported, in which case I could see it accepted at TMLR. I would encourage the authors to pursue this path for a future resubmission.

---

### Review · Reviewer_vkpo · 2023-09-19

**Summary Of Contributions:**

The paper uses the recently introduced GFlowNet framework to interpret existing popular generative models, including VAEs, diffusion models, autoregressive models, normalizing flows, energy-based models and GANs. It not only provides a new perspective to understand the existing methods but also uses this connection to motivate some improvements to the existing methods, which is demonstrated by improved maximum likelihood estimation with trajectory balance consistency (better likelihood and sample quality tradeoff) and improved DDPM (denoising diffusion probabilistic models) training with learnable variance (faster convergence).

**Audience:**

Yes

**Broader Impact Concerns:**

The "Broader Impact Statement" is well-written.

**Claims And Evidence:**

No

**Requested Changes:**

Overall, I quite like the paper but i think the manuscript itself requires some more work to be accepted by TMLR.
I request the author(s) to make the following changes.

## Tuning the presentation for TMLR
TMLR has no page limit and some of the tricks to make the paper fit into 9 pages or so makes it difficult to read.
- 2nd line of 1st paragraph of 5.2: "Figure 2" -> please bring it to the main text
- end of 5.2: "We defer other results and details to Section C.1" -> please bring appendix C.1 to the main text
- 4th line of the 3rd paragraph of 5.3: "Section C" -> please bring appendix C to the main text
- 4th line of the 3rd paragraph of 5.3: "Table 5.3" -> I cannot find table 5.3
- end of 5.3: more discussion of figure 1 should be provided.

Typos
- 4th line of 2nd paragraph of 2.2: "qual" -> "qual to"
- 2nd line of 3rd paragraph of 5.3: "then" -> "than"

## Making the main claim precisely match the paper content
Unlike other models, the connection to GANs is not satisfying in the sense the an alternative algorithm that is impossible/intractable has to be proposed to make the connection.
The current main claim that the paper "Our framework provides a means for unifying training and inference algorithms"/"unifying different kinds of generative models" seems to be a little over-claim in the sense that GANs seem to be a special case that the framework cannot provide a tractable algorithm.
What I mean is that "interpreting the learning of X with an intractable algorithm" sounds trivial and could probably be done by many perspectives that might not be interesting.
For this, I believe that (1) such special case should be made/mentioned clear in advance so that the claim is precise and the main text supports it well and (2) add more discussions on why it is such a special case; current it only stops at "..., which makes this algorithm intractable", which is unsatisfying. Is there some deeper connection that lead to this fact?

**Strengths And Weaknesses:**

## Strengths
The paper connects many generative modeling methods to the GFlowNet framework and provides an insightful discussion on the connection.
I believe it is a useful reading to the community as to understand their connections better and gain some interesting insights regarding RL (section 4).
And as suggested, it is expected to use this perspective to improve existing methods, which could lead to many future work.

## Weaknesses
The presentation of the paper is not ideal.
For one, the paper is not as well organized as one would hope to see such a "connection paper".
I feel it is more like a collection of interesting facts, per se, but I could be subjective on this.
For the other, the paper seems to be originally prepared for a short (8/9) page conference paper and many details are omitted or moved to the appendix.
This has to be changed for an acceptance.

Apart from writing, I think the main claim of the paper should be made clear or updated to reflect the actual discovery in the main text.
Please refer to the "Requested Changes" section for the last two points.

---

### Review · Reviewer_d5zx · 2023-10-10

**Summary Of Contributions:**

This paper describes the relationship between GFlowNets and a broad set of generative models. For example, the hierarchical VAE can be perceived as a GFlowNet in the sense that the generative model corresponds to the forward policy and the inference model the backward policy. Thereby the framework of GFlowNets can be used for further development of improved generative models.

**Audience:**

Yes

**Broader Impact Concerns:**

Papers that are trying to merge concepts, frameworks and methods are important for the machine learning community for a unified development within generative modeling.

**Claims And Evidence:**

Yes

**Requested Changes:**

If this paper had a stronger evaluation, including ablation experiments, on the MLE-GFN compared to other state-of-the-art methods, this paper would be very interesting to the broader machine learning community.

**Strengths And Weaknesses:**

1. The paper provides a clear basis for describing GFlowNets in the context of generative models is interesting.
2. The theoretical arguments are easy to follow and to the reviewer, there are no immediate errors.
3. In order for this paper to be stronger, it would be beneficial to see more results and evaluations of why "GFlowNet acts as a general-purpose glue for tractable inference and training." Without a stronger results section it is not the reviewers opinion that the theoretical finding is strong enough for publication in its own right.

---

### Review · Reviewer_Wk7X · 2023-10-20

**Summary Of Contributions:**

* Authors offer an interpretation of several families of generative models as special cases of the GFlowNet framework.
* Authors propose MLE-GFN, a new method for training GFlowNet-like model in the generative modelling setting (rather than in the sampling setting, as per authors nomenclature outlined in Section 4)

**Audience:**

Yes

**Claims And Evidence:**

No

**Requested Changes:**

Almost all of the weaknesses from the previous section are addressable.

Typos/odd phrasing (in my perception):

* Sec 3.3: “Furthermore, if we P_B(\tau|x) is deterministic”
* Sec 5: “In this section, we investigate that whether”
* Sec 5.1: “Since the gap between the inherent difference between”
* Sec 5.1: the closing “ character after ”TBC“ and ”trajectory balance consistency“
* Sec 5.1: “howbeit” seems really archaic and I did not know what it means at first
* Sec 5.1: “The proposition demonstrates the correctness of optimizing on L_TBC to minimize the divergence.”
* Sec 5.2: “The experiment in this subsection followS”, “7 different distributionS”, and “We consider with the algorithm and baselines [...]”
* Sec 6: “this unification implies a method of constructing an agglomeration of varying types of generative modeling approaches, where GFlowNet acts as a general-purpose glue for tractable inference and training.”

**Strengths And Weaknesses:**

Strengths

1. I found the Sections 3.1, 3.2.1, 3.3 interesting and allowed me to build some high level intuition/understanding of the GFlowNet framework in the context of the methods I’m familiar with and see the possible equivalences. (I lacked familiarity with the GFlowNet literature before reading this work.)



Weaknesses (all fixable, I think)

1. Background on GFlowNets, Section 2.2
    1. Currently this section feels too terse while in my opinion being a crucial for the entire paper to serve its role. I would expect a more accessible/thorough/slower exposition of the existing GFlowNets literature.
    2. No comment is made about the discrete vs continuous data sampling/modelling with GFlowNets, while most of the other methods in Section 3 focus primarily on continuous data. My understanding is that working with continuous data using GFlowNets is only a recent innovation (arXiv preprint 2301.12594).
2. Section 3
    1. While I agree this section allows the reader to relate GFlowNets to some of the other families of generative models, I remain unconvinced about the claim that, in its current form, the work facilitates extensions of other generative methods due to GFlowNet inspirations. See also Weaknesses for Section 5.3.
3. Section 4
    1. I find the nomenclature confusing, although I agree it is not easy to find appropriate terms for what the authors are trying to describe. In my view, the term “generative modelling” encompases all of the methods mentioned in this paper, and I suspect I am not the only one who would say so. Quickly brainstorming, the most accurate terms that come to my mind for the distinction that the authors are trying to make are: “generative models learned from samples” and “generative models learned from the unnormalized distribution”.
    2. The way this section is currently presented, it feels out of place wrt to the narrative of the paper. I don’t entirely understand it’s purpose - how does the message conveyed by this Section fit into the contributions of this paper? Maybe it would serve its purpose better if this distinction was pointed out in Section 2.1? This way the reader would be aware of this aspect while reading Section 3. Not entirely certain how to address this.
    3. What is the difference between “high variance and high stochasticity”?
4. Section 5.1, 5.2 and Appendix C.1
    1. I think the paper currently does a poor job describing the need for MLE-GFN algorithm and its novelty. In the paper’s current shape, this contribution reads and feels like an afterthought needed to get the paper in length for the submission.
    2. Is MLE-GFN the only algorithm/method(/combination of objectives) among the ones mentioned in the paper, that allows for training GFNs using a dataset (empirical measure) rather than access to the unnormalized target density function p*(x)? If not, what other algorithms allow for this? Does EB-GFN also allow for this? Is this the first such algorithm proposed in the literature? After reading the paper, and revisiting multiple fragments multiple times, I still do not know the answers to these questions (I still might have missed it) - I feel these answers should be answered quite explicitly to position the contribution of this paper in the context of the literature.
    3. I consider the name of MLE-GFN a bit of a misnomer given Eqs 19-20 used in Algorithm 1 are a lower bound on evidence, rather than equivalent to maximum likelihood objective.
    4. I am not fully convinced by the evaluation protocol.
        1. I remain unconvinced about the authors’ explanation about why they consider MMD metric more appropriate than the NLL metric in Appendix C.1. I would love to hear other reviewers opinion on this.
        2. Could the authors add a row of samples from EB-GFN to Fig. 2 to showcase the mode collapse they mention?
        3. Does the NLL for the GFlowNets need to be estimated or can it be computed exactly? If the former, how is it estimated?
        4. Maybe if the authors argue that the tasks performance is being saturated, it would be appropriate to find a different benchmark problem, or scale down the capacity of the networks learned to the point of eliciting differences between the two methods?
    5. My impression is that authors seem to try to convince the reader that MLE-GFN outperform EB-GFN. I don’t feel that’s true, neither it is necessary for this work to be published in TMLR. I would rather see a more genuine (in my view) narrative of what is going on.
5. Sections 5.3 and Appendix C.2
    1. In my understanding, the purpose of these experiments is to showcase how the GFlowNets perspective on diffusion models leads to new insights for other generative modelling methods. However I find the way they try to prove this point unconvincing - I don’t think that the GFlowNets perspective yields an insight to learn the Beta values.
    2. In Table 2, I’m putting low confidence to the takeaway that “MLE-GFN outperforms iDDPM" since the differences are rather small and I suspect the baselines’ hyperparameters are not very precisely tuned (since the authors are training those models from scratch in the computationally constrained setting they described).
    3. Overall, I would prefer this paper without this section altogether.
6. Title
    1. I find the term “unifying” far too strong and inaccurate given the content of the paper. I would find something like “Charaterizing the relations between GFlowNets and generative modelling methods” more accurate. I’m not suggesting this title in particular, just brainstorming out loud.
    2. The current title doesn’t really reflect the second contribution of the paper - introducing the MLE-GFN objective.
7. Clarity of writing
    1. Given that, in my opinion, the main value of this paper lies in knowledge dissemination rather than the novelty of its contribution, I would encourage the authors to take an extra editing pass to improve the clarity of writing.

I have not verified the correctness of sections 3.2.2 and 3.2.3.

---

### Decision · Action_Editor_stJh · 2023-11-25

**Recommendation:** Reject

**Comment:**

Overall, reviewers appreciated the effort of connecting the newer GFN model class to older DGMs. However, in their reviews, three reviewers raised several concerns including how precise the claims and presentation of the paper are. In summary, these can be grouped as i) lack of enough background on GFNs, ii) lack of substantiating what advantages the connections between model classes can bring and iii) the experimental part feeling disconnected from the first half of the paper and being limited.

Part ii) is the most crucial, as the proposed generalization do not seem to bring an operational or representational novelty. Their argument could be synthesized with an analogy as follows: we can easily say that every non-latent variable model can be represented as a latent-variable model, or any probabilistic model can be represented as an energy-based model, or every algorithm as a Turing machine, but what do we get? and what do we lose?

During the rebuttal, authors have submitted an improved revision, successfully dealing with concerns related to part i). Still, reviewers judged the other concerns not properly addressed and the new revision not yet focused enough, superficially touching the paragraphs that needed to be rewritten (and the claims amended).

I personally appreciate the contributions, and believe the paper improved in the rebuttal, but at the same time agree with the reviewers that  concerns related to part ii) still need to be addressed. I encourage authors to further and precisely elaborate on what these ``bridges'' between model classes can bring and what is lost in the abstraction. Special care should be taken when distinguishing between model classes and inference algorithms, as highlighted by reviewers `Wk7X` and `mPir`. The presentation and evaluation of the proposed MLE-GFN should be strengthened, by providing more empirical evidence of their benefit. I believe that taking into account all these aspects can turn the paper into a strong contribution.

As such the paper is rejected, but authors are invited to resubmit a major revision that can address the concerns raised by reviewers `vkpo`, `Wk7X` and `mPir`.

**Audience:**

The TMLR audience is definitely interested in generative modeling and a paper surveying the current state of the art in deep generative models while comparing and contrasting them with GFNs would be of certain interest.

**Claims And Evidence:**

The authors revise many classes of deep generative models (DGMs), from GANs to hierarchical VAEs and diffusion models but also autoregressive models, and try to reinterpret them as generative flow networks (GFNs), a recent class of trainable simulators that cast learning to sample as learning trajectorys in an MDP defined over partial (possibly augmented) samples. Furthermore, the authors propose MLE-GFN, a new way to train GFNs inspired by a maximum likelihood principle.

In its original version and in the current one, the manuscript presents GFNs as exact generalizations of the other DGMs (minus GANs, amended in the revision). As noted by the reviewers, these claims are hard to sustain with the current presentation because they are not precise enough as sometimes ideas of how to train a model class or how to represent it are conflated, and some of the generalization proposed are known in the literature. Furthermore, the first part of the paper that introduces such claims should be supported by a newly found operational or representational advantage for the original model classes. The MLE-GFN training procedure is a nice addition for the GFN literature, connected to autoregressive models and normalizing flows.

**Resubmission Of Major Revision:**

The authors may consider submitting a major revision at a later time.